# A Cross Entropy Based Deep Neural Network Model for Road Extraction from Satellite Images

**DOI:** 10.3390/e22050535

**Published:** 2020-05-09

**Authors:** Bowei Shan, Yong Fang

**Affiliations:** School of Information Engineering, Chang’an University, Xi’an 710064, China; bwshan@chd.edu.cn

**Keywords:** cross entropy, encoder-decoder, road extraction, deep convolutional neural network

## Abstract

This paper proposes a deep convolutional neural network model with encoder-decoder architecture to extract road network from satellite images. We employ ResNet-18 and Atrous Spatial Pyramid Pooling technique to trade off between the extraction precision and running time. A modified cross entropy loss function is proposed to train our deep model. A PointRend algorithm is used to recover a smooth, clear and sharp road boundary. The augmentated DeepGlobe dataset is used to train our deep model and the asynchronous training method is applied to accelerate the training process. Five salellite images covering Xiaomu village are taken as input to evaluate our model. The proposed E-Road model has fewer number of parameters and shorter training time. The experiments show E-Road outperforms other state-of-the-art deep models with 5.84% to 59.09% improvement, and can give the accurate predictions for the images with complex environment.

## 1. Introduction

In recent years, the great success of deep learning has influenced many areas. In remote sensing community, many problems, such as, understanding high spatial resolution satellite images, hyperspectral image analysis, SAR images interpretation, multimodal data fusion, and 3-D reconstruction have employed the deep learning technique. Readers may refer to the review papers [1,2] for details. In 2015, Chen et al. [3] presented a Deep Convulutional Neural Network (DCNN), named DeepLab, to address the task of semantic image segmentation, and achieved state-of-the-art results at the PASCAL VOC-2012 segmentation benchmark. Thereafter, a series of improvement to DeepLab has been made [4,5,6,7,8], e.g., DeepLabv1/2/3/3+. The experiemnts of these models have demonstrated that DCNN-based algorithms are powerful tools for semantic image segmentation. This is because of the built-in invariance of DCNNs to local image transformations, which allows them increasingly to learn high level abstract data representations.

Road extraction is to identify and label the roads from satellite land images either in pixel level or with skeletons. This task has been intensively studied in both computer vision and computer graphics community [9,10,11]. Automating generation of road networks has a wide range of applications, e.g., crisis response in remote areas, map updating, city planning, autonomous cars, etc. Although many researchers have paid attention to it, obtaining accurate results automatically is still a challenging task due to the complex backgroud, occlusion and noise in raw satellite imagery.

## 2. Background

In fact, road extraction is a sub-problem of semantic image segmentation, in which only two types of objects, i.e., road or background, are considered. Under this framework, a lot of algorithms have been proposed by taking advangtage of deep learning models. Mnih and Hinton [12] first used the Restricted Boltzmann Machines (RBMs) to learn to detect roads from high spatial resolution aerial images. Zhang et al. [9] proposed a Deep Residual U-Net (DRUnet), which combines the deep residual learning and U-Net architecture for road extraction. Zhou et al. [13] presented the D-LinkNet which is built on LinkNet with dilated convolution layer in the central part. Liu et al. [14] developed a multitask convolutional neural network to simultaneously predict road surfaces, edges, and centerlines from very high spatial resolution remotely sensed images in complex urban scenes. Gao et al. [15] devised a refined deep residual convolutional neural network (RDRCNN) framework with a postprocessing stage for road extraction.

All above studies did not consider the loss of spatial resolution raised by unpooling operations. Our work use Atrous Spatial Pyramid Pooling (ASPP) algorithm to address this problem and apply PointRend technique to get a smooth road boundary with good connectivity. These improvements in turn make the extracted road maps have a more accurate results. Our main architecture is inspired by the image segmentation model, DeepLab-v3+ [8]. We propose an end-to-end learning method, called E-Road, which has an encoder-decoder architecture. At the encoder side, a deep convolutional neural network (DCNN) classifies the pixels of one satellite image into two subsets: either road or background. The decoder generates the sharp boundaries and gradually recovers the spatial information of the road. Our contributions can be summarized as:1)We present a novel encoder-decoder deep network which employed a ResNet as encoder modual and a simple yet effective upsampling layers and PointRend algorithm as decoder module.2)We use an Atrous Spatial Pyramid Pooling (ASPP) technique to trade off between precision and running time.3)We apply a modified cross entropy loss function to enhance the performance of training process for road dataset.4)We employ an asynchronous training method to speedup the training time without loss of performance.5)Our proposed model achieves the excellent performance with less network comlexity compared with other deep networks.

## 3. Methods Description

### 3.1. Encode-Decoder Architecture

We borrow the idea from DeapLabV3+ [8] to construct E-Road architechture, which can be divided into two parts: encoder and decoder. The encoder consists of many layers of convolutional neural networks, which takes satellite images as input, aggregate features of images at multiple layers, and extracts dense feature to generate high dimensional feature vectors. The decoder takes high dimensional feature vectors as input to generate road network. The road extraction problem is partly different from normal semantic image segmentation. The satellite images may have very rich information of the details, while the extracted road masks have very poor semantics, i.e., road or not road. To address this task, we modify the encoder of DeapLabV3+ to a more shallow networks to well preserve the details of images. The networks architecture of E-Road is depicted in Figure 1.

The raw satellite images are cropped into the size of 512 × 512 with 3 channels as the input. The backbone the encoder is a ResNet [16] network, which uses residual blocks and downsampling blocks at each layers. To investigate the performance influnced by deep model, we use two ResNet: ResNet-18 with 18 layers and ResNet-34 with 34 layers respectively in the experiment. ResNet network makes our model achieves the same performance with the VGG network for object classification task, with fewer filters and lower computing complexity. Comparing VGG-19 with 19.6 billion FLOPS operations (multiply-adds), ResNet-18 has only 2.1 billion FLOPS, and ResNet-34 has only billion FLOPS. The detailed network architectures of two ResNets are illustrated in Figure 2. The contextural information at multiple scales is acquired by Atrous Spatial Pyramid Pooling (ASPP), which will be introduced at the next subsection.

Inspired by the noval idea of [17], the decoder consists of two parts: upsampling (by 4 times) layers and PointRend which will recovery the learned features to detailed road boundary. The PointRend module accepts different layers of upsampled CNN feature maps as input and outputs the predicted road maps with higher spatial resolution. PointRend first carefully selects some ambiguous points on the boundary of the segments. Then it conduct a extraction of point-wise feature representation based on selected point by interpolation. The output labels are predicted by a point head network from the point-wise features. The detailed implementation of PointRend is described in Section 4.

### 3.2. Atrous Spatial Pyramid Pooling

Inspired by [7], we advocate the ASPP to reduce the spatial resolution of the resulting feature maps, instead of repeated combination of max-pooling and striding. Recently, [18] presented a novel algorithm named Waterfall Atrous Spatial Pooling (WASP), which consists several branches of atrous convolution in a waterfall-like configuration. WASP could improve the performance with less number of parameters and memory required. While WASP is more complex than ASPP to be implemented due to the connections between the neighbouring atrous convolution modules, and the parallel architecture makes ASPP much easier be extended with more atrous convolution layers with larger rates. In addition, our work use a light-weight backbone ResNet-18 which can greatly reduce the parameters and memory, hence we would rather apply ASPP than WSAP. The two-dimmensional Atrous Convolution is defined as follows.

For 1D input signals *u*, the output *v* of atrous convolution at location *i* with a filter f[k] of length *K*, is computed by:(1)$$v\left[i\right]=\sum_{k=1}^{K}u\left[i+r\cdot k\right]f\left[k\right],$$
where the atrous rate *r* corresponds to the stride rate. If rate r=1, atrous convolution goes back to standard convolution. We can use atrous convolution to explicitly control how densely feature responses are computed in ResNet-18. Three parallel 3×3 convolution branches (rates = 6, 12, 18 respectively) are employed, which is illustrated in (i) of Figure 1. As mentioned in [7], if the atrous rate goes to a larger value, the number of effective filter weights becomes smaller and invalid. To address this problem, an image pooling, which is illustrated in (ii) of Figure 1, is applied on the last feature map to incorporate global context information.

### 3.3. Sigmoid Function

Let $z_{i}$ be the output of two upsampling layers. The sigmoid function is performed to map the estimated output values $y_{i}$ into (0,1):(2)$$y_{i}=\frac{1}{1+e^{-z_{i}}}.$$

### 3.4. Modified Cross Entropy Loss Function

The road extraction can be modeled as a binary classification problem, in which one pixel either belongs to the road or to the background. Normally, the cross entropy loss function of binary classification is defined as:(3)$$L=-\frac{1}{n}\sum_{k=1}^{n}\left(t_{i}\log y_{i}+\left(1-t_{i}\right)\log\left(1-y_{i}\right)\right),$$
where $t_{i}$ is the groundtruth of the *i*th pixel. $t_{i}=0$ represents the *i*th pixel belongs to background and $t_{i}=1$ represents it belongs to road. $y_{i}\in \left(0,1\right)$ is the estimated value of the *i*th pixel after sigmoid function. As $y_{i}$ approaches 1, the *i*th pixel more likely belongs to the road. Our training process minimizes loss function *L* by iteratively adjusting the weights of network.

Equation (Equation 3) has two shortcomings: First, all pixels play the same role to evaluate the loss function, which could ignore the special location information of $t_{i}$; Second, this loss function is more suitable for the case of balancing positive/negative examples, while most road extraction datasets may not satisfy this requirement. Our dataset, DeepGlobal [19], is split to training/testing/validation subsets. There are 4.5% positive and 95.5% negative pixels in the training dataset, 4.1% positive and 95.9% negative pixels in the test dataset, and 3% positive and 97% negative pixels in the validation dataset. Considering these two concerns, we redesign a modified cross entropy loss function, by regarding the influence of pixels spacial location and the serious unbalanced positive/negative examples. We first define a function $g(l_{i})$:(4)$$g\left(l_{i}\right)=\left\{\begin{array}{ccc} 0 & l_{i} & =0\\ \frac{l_{i}}{\max_{j\in I}\left\{l_{j}\right\}} & 0 & <l_{i}<T\\ \frac{T}{\max_{j\in I}\left\{l_{j}\right\}} & l_{i} & >T \end{array}\right.$$
where $l_{i}$ is the Euclid distance between *i*th pixel and nearest road, which needs be computed from examples before training. $T=0.3\max_{j\in I}\left\{l_{i}\right\}$ [20] is a therehold to determine whether the pixel is far enough from the road.

Thereafter, the modifed loss function is defined as:(5)$$L=-\frac{1}{n}\sum_{i=1}^{n}\left(\alpha_{1}t_{i}\log y_{i}+\alpha_{2}e^{-g(l_{i})}\left(1-t_{i}\right)\log\left(1-y_{i}\right)\right),$$
where $\alpha_{1}$ and $\alpha_{2}$ are defined as [21]:(6)$$\alpha_{1}=\frac{N_{n}}{N_{p}+N_{n}},\alpha_{2}=\frac{N_{p}}{N_{p}+N_{n}}.$$
where $N_{p}$ and $N_{n}$ are the example numbers of positive and negative respectively.

Equation (Equation 5) takes into account the influence of road continuity, and the different weights of positive/negative examples in the loss function could accelerate the training process.

## 4. PointRend Algorithm

Current road extraction deep models [9,10,11,12,13,14,15] focus on recovering the contour information and connectivity of the road map, and none of them pays attention to the road boundry smoothness. Recently, Kirillov et al. [22] presented a PointRend technique, which can obtain a smooth and sharp boundry by rendering method in the image segmentation process. Inspired by this noval idea, in this section, we integrate the Point-based rendering algorithm into the E-Road deep model.

Abstractly, PointRend algorithm takes the decoder feature maps $f\in R^{C\times H\times W}$ as input over a regular grid, where *C* is the channels number, and *H* and *W* are the height and width of the maps. PointRend outputs the predicitons for the two class labels $l\in R^{2\times H\prime \times W\prime }$ over a regular grid of higher spatial resolution. There are three modules in the PointRend: (i) point selection, (ii) point-wise feature extraction, and (iii) point head. It should be noted that PointRend is incorporated but not limited to the E-Road model. It can be applied to any CNN based image semantic segmentation task to handle the coarse-to-fine object boundaries problem in an anti-aliasing fashion. The architecture of the PointRend algorithm is illustrated in the Decoder part of Figure 1.

**Point selection** is an adaptive subdivision algorithm. During it, we iteratively render the output image with a coarse-to-fine manner. The first prediction is the coarsest, and is performed on the point of regular grid. In the following iteration, the previously predicted segmentation is upsampled by a bilinear interpolation, and then on this denser grid, the *N* most uncertain points $n_{i}^{*}\left(i=1,2,\ldots ,N\right)$ are selected by Equation (Equation 7).
(7)$$n_{i}^{*}=\arg\min_{n_{i}}\left|p\left(n_{i}\right)-0.5\right|.$$
where $p(n_{i})$ is the probability for point $n_{i}$ belonging to a binry mask. Once *N* points are selected, a point-wise feature extraction is performed.

**Point-wise features** are constructed by concatenating two types of features, i.e., coarse predicted features and fine-grained features on the selected *N* points. (i) The coarse predicted features is a 2-dimensional vector at each point in the region, which represents a 2-class prediction. The coarse prediction conveys more globalized and general context, and the channels provide the semantic classes. For E-Road, it can be predicted from a stride 18 feature map. This predictions are similar to the outputs made by the existing architectures. (ii) The fine-grained features is a vector containing the fine detailed segmentations, which is extracted by performing a bilinear interpolation each sampled points from deep feature maps. PointRend uses a Hypercolumn method [23] to concatenate the features extracted from multiple feature layers, e.g., Res2, Res3 … in ResNet.

**The point head** is a neural network with small size, which predict lables based on above point-wise features. This neural network is a Multi-layer Perceptron (MLP). Our MLP has three hidden layers, in which the 256 output channels with two coasrse predictin features are supplied to make the input vector for the next layer. The ReLU is used in the MLP and Sigmoid function is applied to its output. Similar to the graph convolution [17], the MLP shares weights acroos all regions and all points.

Above process is iteratelly performed until the desired spatial resolution is achieved by upsampling. We illustrate one step of this process in Figure 3 Let our desired output spatial resolution be R×R pixels, the initial spatial resolution be $R_{0}\times R_{0}$ spatial resolution and the number of point predictions be $N_{P}$. It’s obvious that:(8)$$N_{P}\leq N\log_{2}\frac{R}{R_{0}}$$
where *N* is the number of the selected points. Equation (Equation 8) allows PointRend to perform super-resolution prediction with less computing complexity.

**Training of PointRend.** In the training process of PointRend, the point head needs to select points to train MLP. Generally, the adaptive subdivision algorithm can be used as selection strategy. While, the sequential process in this strategy is not suitable for the backpropagation of neural network. We will use a random sampling based selection strategy instead.

The random sampling strategy bias selects on the whole maps with some degree of uniform way. Let *k* be the upper parameter with (k>1). Let β be the lower parameter with (0≤β≤1). The selection strategy has three steps: (i) *Randomly sampling*. We generate much more candidate points by randomly sampling kN points from a uniform distribution (ii) *Boundary sampling*. For the uncertain area, such as road boundary, we interpolate the coarse predictin values at all kN points. The the most uncertain βN points are choosen from kN candidate points and are used to calculate a task specific uncertainty estimate. (iii) *Rest sampling*. The non-boundary area sampled by the arest (1−β)N points from a uniform distribution as well. This prececue is illustrated in Figure 4. It is found that with the increase of *k* and β the distribution of points turns from uniform in whole map to heavily biased to the boundary (uncertain area). In addtion, the computing burden also increased dramatically. To make a trade-off between precision and training complexity, we takes mildly biased sampling strategy, and set $N=16^{2},k=4.5,\beta =0.8$.

## 5. Experiments

Our model is trained on the DeepGlobal datasets, and is evaluated on several satellite images located in Xi’an city, P.R. China. We implement our deep learning model on Tensorflow [24] framework, and train it on one NVIDIA TITAN V GPU.

### 5.1. Dataset

The DeepGlobe dataset [19] consists of 8570 images and spans land area of 2220 km${}^{2}$, which is captured over Thailand, Indonisia, and India. The ground spatial resolution of the image is 50 cm/pixel with three channels (Red, Green and Blue). Among them, 6226 images (72.65%) were chosen as the training dataset. In total, 1243 images (14.50%) were chosen as the testing dataset and 1101 images (12.85%) were chosen as the validation dataset. Each images has a size of 1024×1024. To meet the input of our deep learning networks, we split each one into images with a size of 512×512; therefore, our new DeepGlobe dataset has a total 34,280 images. This augmentation of dataset could reduce the risk of overfitting on the training process.

### 5.2. Evaluation Metric

Road map extraction is a binary classfication problem. Given label and prediction, let TP be true-positive; TN be true-negative; FP be false-positive; FN be false-negative. Based on the above definition, we use F1 score, recall, overall accuracy (OA) and pixel-wise Intersetion over Union (IoU) as our evaluation metric. F1 score is a metric for the harmonic mean of precision and correctness, which can be computed as follows:(9)$$F1=\frac{2TP}{2TP+FP+FN}.$$

Recall is used to determine how many relevant pixels are correctly predicted and is defined as follows:(10)$$recall=\frac{TP}{TP+FN}$$

OA gives the precision of predicted roads and background and can be calculated as follows:(11)$$OA=\frac{TP+TN}{TP+FP+TN+TN}$$

IoU is the ratio of the overlapping area of predicted pixels and groudtruth pixels to the total area. In our road extraction task, it can be defined as follows:(12)$$IoU=\frac{TP}{TP+FP+FN}$$

IoU can only evaluate the model for one image. If one wants to evaluate a dataset *D* with more than one images, the mIoU metric can be used:(13)$$mIoU=\frac{1}{\left|D\right|}\sum_{i\in D}IoU_{i},$$
where $IoU_{i}$ is the IoU of the *i*th image, and |D| is the image number of dataset *D*.

### 5.3. Training Process

We use Equation (Equation 5) as the loss function and Adam [25] as the optimizer. We borrow the idea of a “poly” learning rate policy [4] in our work, and $learning\;rate=initial\;learning\;rate\times {\left(1-\frac{iter}{max\_iter}\right)}^{power}$, where initiallearningrate=2e−4, and power=0.9. The batch size in our training process is fixed as 8. We train and validate our model on DeepGlobe dataset with 120 epochs and plot the loss value in Figure 5. It is found that at first several epochs errors drop abruptly, and after 25 epochs both training and validation errors converge to the minimum values. To avoid the possibility of overfitting, we set the training epoch as 40.

Our augmented DeepGlobe dataset has 24,904 images for training. To accelerate the training process, we apply the Tensorflow asynchronous training technique [24] in our experiment, which means that each replica of the training loop executes independently without coordination. The training times are plotted in Figure 6 under different replicas. To investigate the performance of asynchronous training, we also plot the mIoU for DeepGlobe validation dataset under different replicas in Figure 7. It can be found that more replicas could significantly reduce the training time almost without performance loss.

### 5.4. Results

To study the performance of our trained model, we take three rural satellite images that cover the Xiaomu village in Shannxi province, China, to perform the evaluation. All three satellite image have the same spatial resolution 1.5 m/pixel. Xiaomu village has some rural roads with farmland, woods, pedestrians and buildings. All these elements have been covered in DeepGlobe dataset. To better indentify the roads from the other objects, we outline the road boundary with blue line in all satellite images. We compare our method with DRUnet [9] and D-LinkNet [13] to investigate the feasibility. In the predicted images, roads are labelled with white pixels, non-road elements are labelled with black pixels. We depict the results via different models in Figure 8. Comparing with the ground truth in Figure 8b, we can find that all three deep models have extracted the road networks from the satellite images. DRUnet and D-LinkNet appear to predict some wrong pixels and could not well keep the road connectivity. While our proposed E-Road has successfully predicted most pixels in road networks and most road connectivities are well preserved. We apply Equations (Equation 9)–(Equation 12) on predicted resutls in Figure 8 to quantitively evaluted the accuracy of different deep models. The different metrics, i.e., F1 score, recall, OA, and IoUs are listed in Table 1. We can find that all three models have demonstrated accurate road extraction ability, while E-Road achieves the highest value in all four metrics. All metrics of E-Road are above 80% and some are above 90%. Compared with the other deep models, E-Road achieves the highest (59.09%) and the lowest (5.84%) improvement. For Image 1 and Image 2, roads are clearly displayed in the satellite images and all three models achieve relative high value. For Image 3, because many buildings and woods have casted shadows on the roads, four metrics decrease and roads connectivity is poor under the prediction of DRUnet and D-LinkNet, while our E-Road still obtains very high metrics for Image 3 with good road connectivities.

To acquire a thorough understanding of our proposed model, we also take additional 10 images covering Xiaomu village as input. The experimental results show that in these 10 images E-Road also outperforms DRUnet and D-LinkNet, and the obtained methrics are between the Image 1 to Image 3. Due to the limited space, we only demonstrate the best and the worst results (Images 1–3) in our paper.

## 6. Discussion

### 6.1. Effects of Depth

The depth of the convolution layers plays a very important role in extracting the high dimensional features from road images. To investigate the effects of network depth, we use two deep networks, i.e., ResNet-18, and ResNet-34 [16] as the backbones of encoder for comparison. We denote them as E-Road18 and E-Road34 respectively, which have the same architecture except with different number of layers. We perform the same training and extraction process as Section 5. The experiment results are depicted in Table 2, where the first row is the loss error of training after 40 epochs, the second row is the training time with 32 replicas and the last row is the IoUs of Image3.

The experimental results show that deep model slightly outperform the shallow one in terms of both loss error and IoUs. While the improvement is not significant, only 2.24% for loss error and 0.60% for IoUs. In addtion, E-Road34 needs much more training time, about 8.47 times, than E-Road18. To make the best trade off between performance and training time, we would take ResNet-18 as the backbone of our deep network.

### 6.2. Effects of PointRend

To investigate the effects of PointRend algorithm, we construct two encoder-decoder models: the decoder of the first one has only upsampling layers and the decoder of the second one has both upsampling layers and PointRend module, and name them E-Road-noPR and E-Road respectively. Both models are trained by the same process as in Section 5.3. We also take two images (named Image 4 and Image 5) covering Xiaomu village with roads, farmlands and buildings. Image 4 and Image 5 have the same spatial resolution of 0.3 m/pixel, which is much higher than Images 1–3. Higer spatial resolution could make the road be more salient. The metrics of the extracted road maps by E-Road-noPR and E-Road are shown in Table 3. We can find that both E-Road-noPR and E-Road achieve almost the same value in both for metrics, and E-Road slightly outperforms E-Road-noPR model. The extracted road maps are depicted in Figure 9. The predicted maps by both models have successfully recovered the contour and connectivity information of the road. Nevertheless, the boundaries of the road by E-Road obviously appear to be much smoother and clearer than that by E-Road-noPR. Therefore, all our results in Figure 8 and Table 1 are obtained by E-Road.

## 7. Conclusions

In this paper, We present a deep convolutional neural network model, E-Road, for road extraction from satellite images. Our model has an encoder-decoder architecture. The encoder employs ResNet-18 network and Atrous Spatial Pyramid Pooling (ASPP) technique, and the decoder uses upsampling layer and PointRend algorithm to recovery road boundary. We train our model via modified cross entropy loss function and asynchronous training technique under augmented DeepGlobe dataset. The experimental results show that our E-Road outperforms the other deep model, and achieves the highest 59.09%, the lowest 5.84% improvement in terms of F1 score, recall, OA and IoUs metrics. The ResNet-18 backbone network and ASPP algorithm greatly reduce the number of parameters in our deep model without loss of performance, and the asynchronous training technique significantly speedups the training process. The PointRend method makes the extracted road has a smooth and sharp boundary with better connectivities. Even for the satellite images with complex environment, our model can also obtain an accurate prediction.

In the future, some additional issues still need to be considered. In our test, the salellite images of Xiaoxu village has a fixed spatial resolution (1.5 m/pixel and 0.3 m/pixel). With the development of remote sensing technology, much higher spatial resolution of imagery would be available. How to adjust the parameters of networks and ASPP to extract the details of road information is a major challenge.

## Figures and Tables

**Figure 1 entropy-22-00535-f001:**
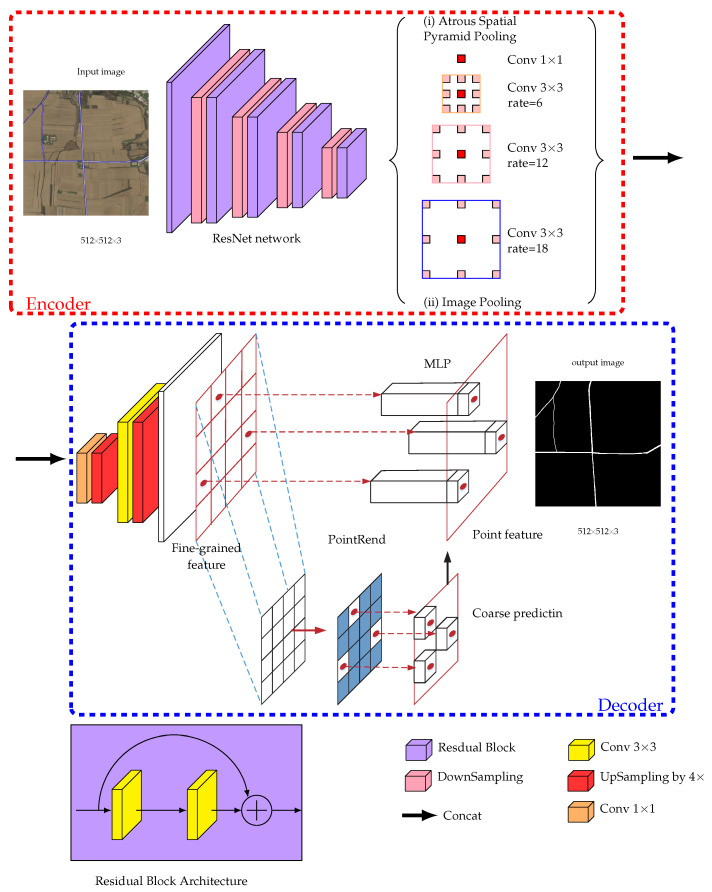
Network architecture of E-Road.

**Figure 2 entropy-22-00535-f002:**
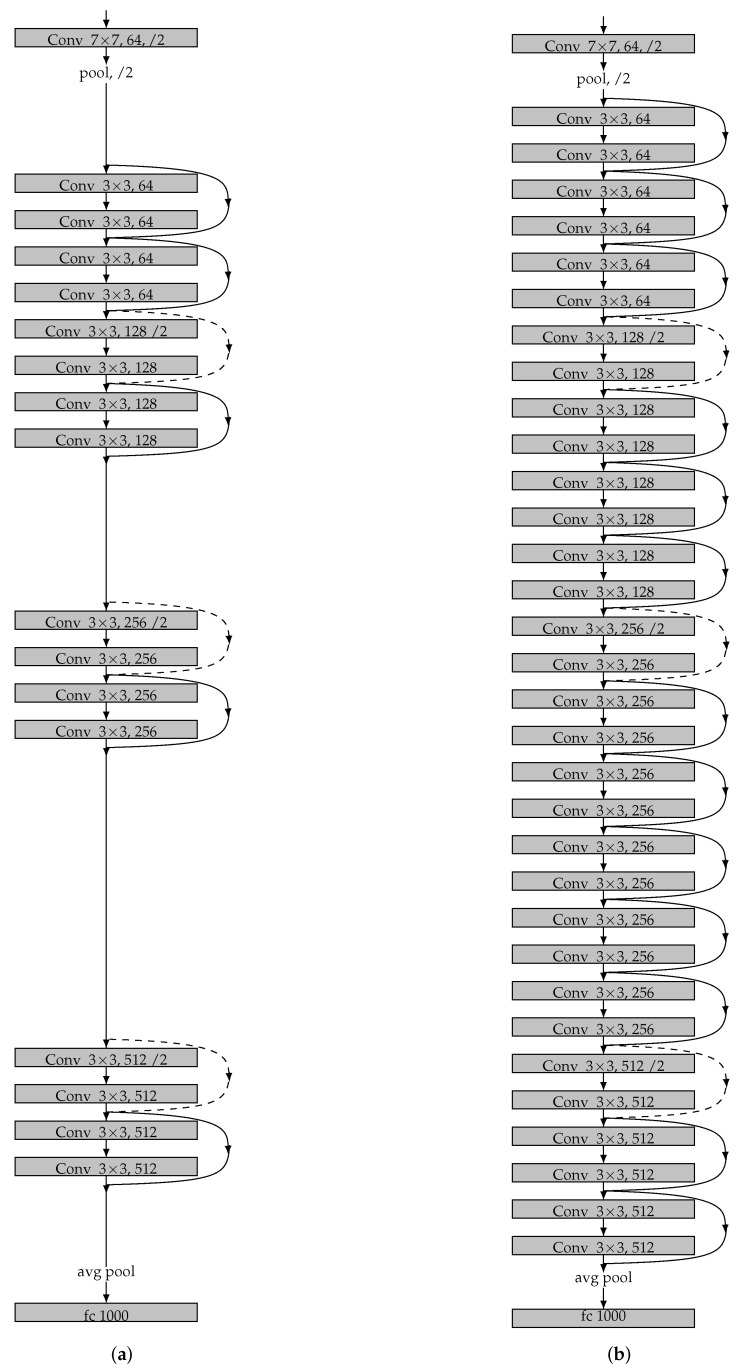
Network architectures of ResNet. (**a**) 18 parameter layers with 2.1 billion FLOPS operations. (**b**) 34 parameter layers with 3.6 billion FLOPS operations.

**Figure 3 entropy-22-00535-f003:**
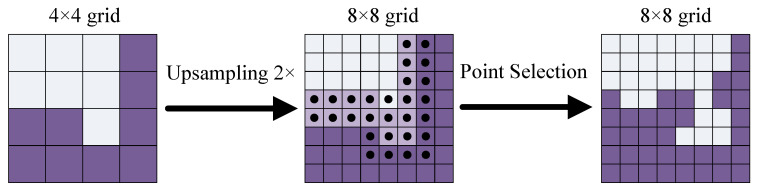
**One Step of PointRend Process**. **Left**: On 4 × 4 grid, a bilinear interpolation is performed to upsample by 2× on the prediction. **Middle**: 28 most ambiguouse points are selected on the finer 8 × 8 grid. **Right**: The detailed point-wise feature are recovered. This process is repeated until the segmentaion is upsampled to the desired spatial resolution.

**Figure 4 entropy-22-00535-f004:**
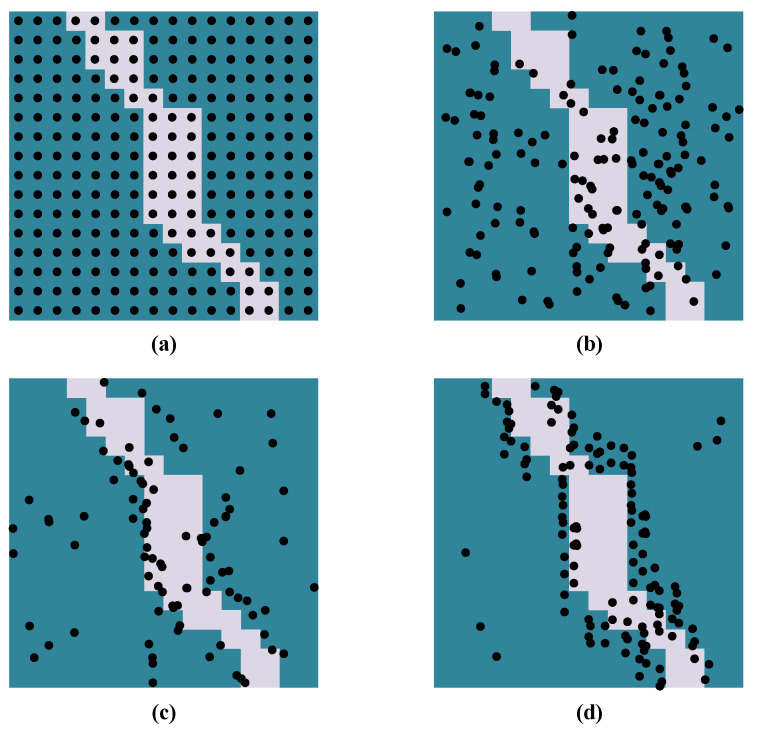
**Training of PointRend**. (**a**): Coarse regular grid. (**b**): points with uniform distribution, k=1, β=0.0. (**c**): Points with mildly biased distribution, k=4.5,β=0.8. (**d**): Points with heavily biased distribution, k=8,β=0.9.

**Figure 5 entropy-22-00535-f005:**
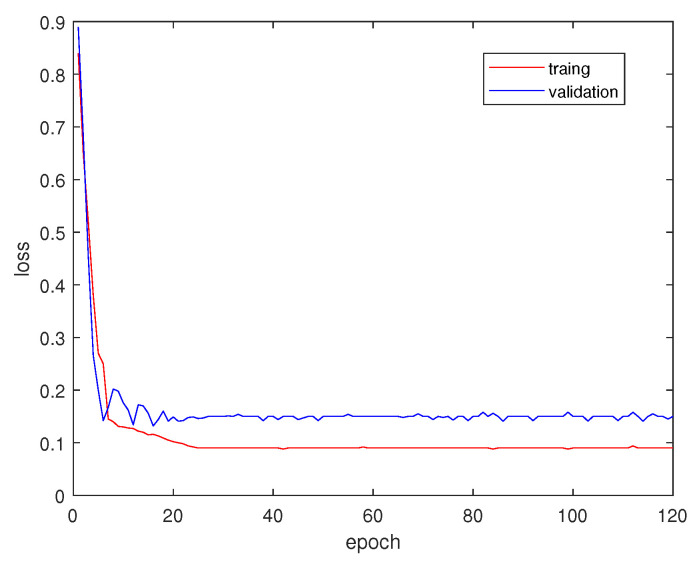
Loss value of traing and validation from DeepGlobe dataset.

**Figure 6 entropy-22-00535-f006:**
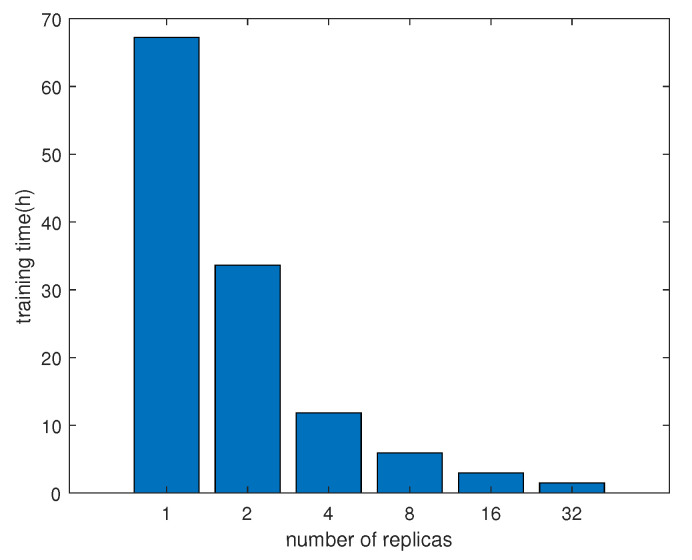
Training time under different replicas.

**Figure 7 entropy-22-00535-f007:**
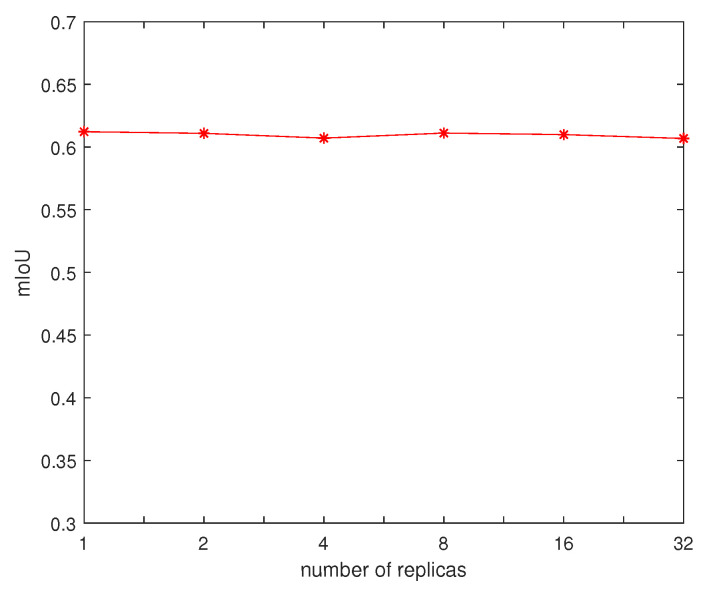
The mIoU under different replicas.

**Figure 8 entropy-22-00535-f008:**
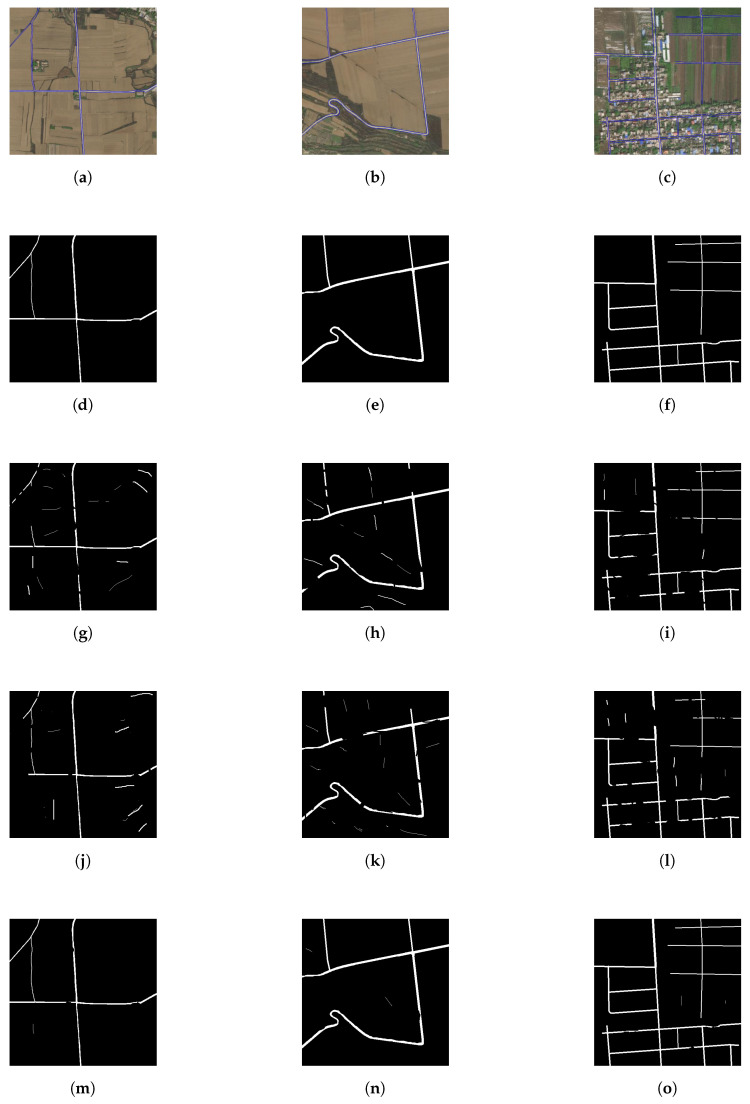
Extracted road networks by different models with spatial resolution 1.5 m/pixel. (**a**–**c**) Input images 1–3. (**d**–**f**) Ground truth 1–3. (**g**–**i**) DRUnet [9] 1–3. (**j**–**l**) D-LinkNet [13] 1–3. (**m**–**o**) Our proposed E-Road 1–3.

**Figure 9 entropy-22-00535-f009:**
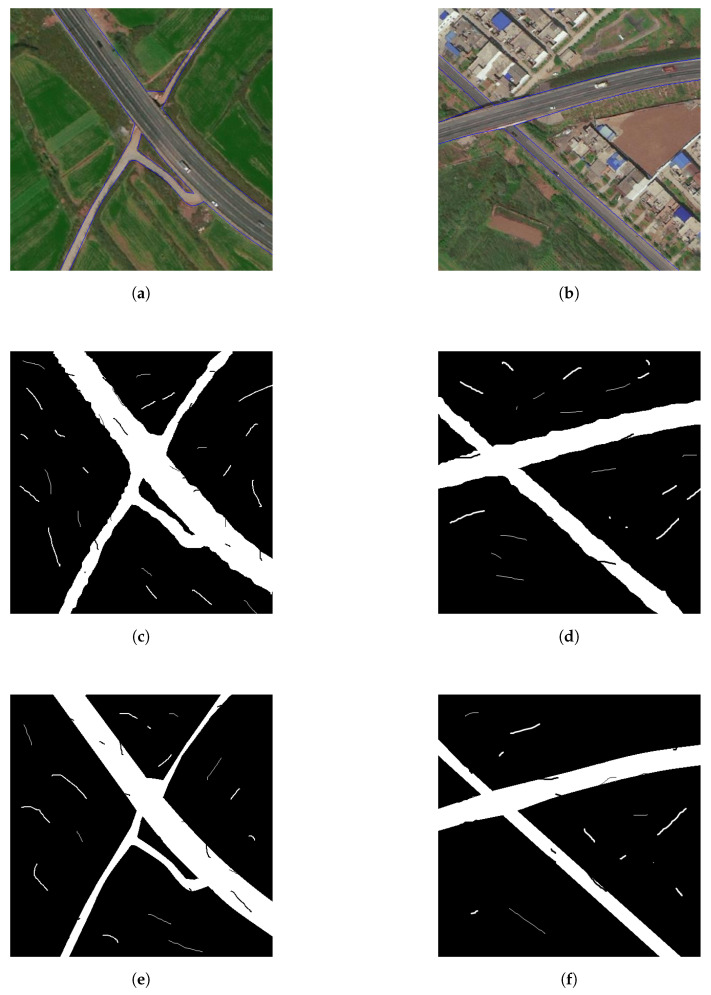
Effects of PointRend with spatial resolution 0.3 m/pixel. (**a**) Satellite image 4. (**b**) Satellite image 5. (**c**) Extracted road map of image 4 by E-Road-noPR. (**d**) Extracted road map of image 5 by E-Road-noPR. (**e**) Extracted road map of image 4 by E-Road. (**f**) Extracted road map of image 5 by E-Road.

**Table 1 entropy-22-00535-t001:** Metrics of extracted three images via different models.

		DRUnet	D-LinkNet	E-Road
Image 1	**F1(%)**	75.53	74.09	**90.86**
	**recall(%)**	67.87	67.20	**86.72**
	**OA(%)**	91.81	91.43	**97.15**
	**IoU(%)**	70.68	78.85	**83.25**
Image 2	**F1(%)**	80.13	79.07	**91.81**
	**recall(%)**	73.53	73.09	**88.27**
	**OA(%)**	91.34	90.97	**96.67**
	**IoU(%)**	66.84	65.38	**84.85**
Image 3	**F1(%)**	63.37	63.04	**93.34**
	**recall(%)**	62.17	62.39	**95.13**
	**OA(%)**	60.50	60.19	**96.25**
	**IoU(%)**	61.49	61.01	**87.51**

**Table 2 entropy-22-00535-t002:** Comparison of two deep networks.

	E-Road18	E-Road34
Loss error	0.089	0.087
Training time (h)	1.5	12.7
IoU(%)	98.82	98.23

**Table 3 entropy-22-00535-t003:** Effects of PointRend.

		E-Road-noPR	E-Road
Image4	**F1(%)**	91.24	92.09
	**recall(%)**	88.65	88.28
	**OA(%)**	98.23	98.43
	**IoU(%)**	91.89	99.05
Image5	**F1(%)**	94.23	94.77
	**recall(%)**	91.53	94.09
	**OA(%)**	97.34	97.97
	**IoU(%)**	93.54	98.58

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
