# Peer review of "A Cross Entropy Based Deep Neural Network Model for Road Extraction from Satellite Images"

_entropy, 2020, doi:10.3390/e22050535_

Round 1
Reviewer 1 Report
This study interesting and the methods are very well explained. However, the authors need to clarify the gap, justify their study, demonstrate their contribution, show their initial and classified images. In addition, a thorough proofreading is necessary.
Please see my detailed comments below.
Please indicate what is the novelty of your work in comparison with the following paper:
Artacho, B. and Savakis, A., 2019. Waterfall Atrous Spatial Pooling Architecture for Efficient Semantic Segmentation. Sensors, 19(24), p.5361.https://arxiv.org/pdf/1912.03183.pdf
Line 14: please change “Tutorial papers” to “Review papers”
Line 26: Please proofread “have pay attention” as it is not grammatically correct.
While you have listed the literature on road extraction from deep learning algorithms in lines 30-38, there is no critic on these studies so that the reader can understand the gap. Please clarify the gap.
It is valuable to justify your work by indication of the gap and demonstration of the effectiveness of your method to fill such gap.
Line 50: please change the verb “employs” to “employed”. This verb is not grammatically correct. I ask the authors to comprehensively proofread the paper and from this part, I will concentrate on comments regarding the content of the research.
Line 69: why the authors have chosen the size of 512*512 and three channels as input?
In Figure 1, sigmoid is not demonstrated while it is mentioned in the legend.
Please refer to the equation numbers in the text.
Please explain how your redesigned function (equation 4) can overcome the two problems you have listed before.
The major problem of this paper is that the classified images are not demonstrated. While it is indicated that three images are used for this paper, there is no demonstration of the images and the segmented roads within these images. It is crucial to see the images first and know how much the complexity of classification for these images in terms of the existence of occlusions among different classes of objects is. Please add the images, results and discuss these details.
Line 225: what the authors mean by “deeper is better”? the differences of the results for IoU with different depth is not considerable and I doubt we can make such conclusion unless we obtain consistent results for more images and after testing several depth levels.
The authors indicate that they have used three images for the classification; however, they talk about image 4 and 5 in line 234. Please elaborate on this.
Reviewer 2 Report
This research developed a new method to extract road network from satellite images based a deep convolutional neural network model with encoder-decoder architecture. The authors also used ResNet-18 and Atrous Spatial 3 Pyramid Pooling technique to balance extraction precision and computing time. To train the deep convolutional neural network model, the authors used a modified cross entropy loss function. A PointRend algorithm was used to recover distinct road boundaries. Research results show that the proposed method outperforms most of the state-of-the-art methods. The reviewer believes that the current version of the manuscript is not yet ready for publication; the authors are encouraged to consider the following comments and suggestions and revise the manuscript accordingly.
- The authors should split the Introduction section into two sections, including Introduction and Background section. The Introduction section should focus on introducing the research objectives and research questions, while the Background section should focus on reviewing of related work and defining the research gap. In addition, the reviewer thinks this paper is more appropriate for another journal such as Remote Sensing.
- The authors should use more generally accepted terms in remote sensing for this paper. For example, in the remote sensing field, there are four types of resolutions, including spatial resolution, spectral resolution, radiometric resolution, and temporal resolution. The authors just mentioned resolution in the manuscript, but they did not discuss it in a specific resolution type. The reviewer assumes the authors are discussing this in spatial resolution, but the authors need to define this explicitly. For example, on page 1, lines 11 and 12, the authors stated that “In remote sensing community, many problems, such as, understanding high-resolution satellite images….”
- For the proposed method, do the input satellite images need to be orthocorrected? In addition, the proposed method uses natural color satellite images (visible blue, green, and red bands) as the input, do additional bands such as near-infrared bands or thermal infrared-band improve the results?
- What is the worst accuracy of the proposed method? In other words, this has to be tested and use the results to further improve the proposed method. A few more input satellite images should be used to test the proposed method to test its consistent accuracy.
- What is the performance of the proposed method in a busy urban environment? A lot of features such as paved parking lot, neighborhood local roads have similar spectral characteristics as the major artery roads. How to separate them in the extracted road network?
- What are the costs of deploying the proposed system in real operational scenario? The authors need to justify this with real dollar amount. Is it cost-effective when processing hundreds of thousands of images?
Reviewer 3 Report
The authors present a novel deep learning framework for the segmentation of remote sensing images. The presentation of the context is clearly explained in the Introduction. Additionaly, the main contributions of this manuscript are also higlighsted in this section, namely: the use of an encoder-decoder network architecture, the use of Atrous Spatial Pyramid Pooling, and the definition of a custom loss function. The proposed method is also correctly presented, and the figures are really appropiate, facilitating the readability. Some comments about the manuscript:
- You are dealing with a segmentation problem. Why did the authors modified the cross entropy loss function, instead of using a more specialized loss for segmentation, e.g. soft-DICE loss?
- In Section 3.1.Dataset, it would be useful to have the percentages of images used for training, validation and test.
- In Section 3.1 Dataset, the authors stated the number of test images to be 1,243. Nevertheless, in the results section (3.4), only the quantitative results from 3 images are shown. Please, provide a table similar to Table 2, where the results of the different models within the test set are shown.
- It would be useful for readers to see some segmentation results (heat maps) in Section 3.
- It is not clear which data is used to perform the experimentation shown in Section 4. Please, provide some additional information in the document to clarify this fact.
- The conclusion section is too short. Please, provide deeper remarking comments about your research in this section.
Other minor comments:
- It would be useful to have more detailed information about the results in the abstract.
- The first section , Introduction, should be numbered as "0".
According to my comments, I think the manuscript should be revised by the authors prior before publication.
Reviewer 4 Report
Dear authors
The article is fluent, the references are updated and proper, the introduction clearly outlines the research problem and the overview on related work is adequate, the methodology is well described and the results are clearly presented.
I would like to suggest a few minor revisions that I hope will be considered.
Line 50. We employs—> we employ
Line 52 Compared other —> compared with other
Line 43-50 The language style of this list should be improved to make it more fluent. “We present” and “we employ” from point 1 and 4 are personal form, point 2 and 3 are impersonal form, also number five can be rewritten.
Line 54-57. The article is well organised so the description of paper organisation is redundant,
Table 1 can be renamed as figure and the three columns can be identified adding “images 1, 2 and 3”
In the captions of table 1 and figure 8 the resolution of the imaged used could be recalled and , if it is possible, add the scale bar.
Best regards
Round 2
Reviewer 1 Report
The paper is significantly improved and a final check of proofreading is recommended.
Reviewer 2 Report
The authors have addressed all my comments.
Reviewer 3 Report
The authors addressed all my comments from the previous review round. From my point of view, the paper is now suitable for publication.
Only one small detail. I commit a mistake in my last review, the Introduction section should be numbered as '1', not as '0'.